# Predicting Colorectal Cancer Occurrence in IBD

**DOI:** 10.3390/cancers13122908

**Published:** 2021-06-10

**Authors:** Mehmet Yalchin, Ann-Marie Baker, Trevor A. Graham, Ailsa Hart

**Affiliations:** 1Inflammatory Bowel Disease Department, St. Mark’s Hospital, Watford R.d., Harrow HA1 3UJ, UK; 2Centre for Genomics and Computational Biology, Barts Cancer Institute, Barts and the London School of Medicine and Dentistry, Queen Mary University of London, Charterhouse S.q., London EC1M 6BQ, UK; a.m.c.baker@qmul.ac.uk (A.-M.B.); t.graham@qmul.ac.uk (T.A.G.)

**Keywords:** inflammatory bowel disease, cancer risk, risk factors, molecular risk factors, molecular biomarkers

## Abstract

**Simple Summary:**

Patients with inflammatory bowel disease are at an increased risk of developing colorectal cancer, and so are enrolled in a surveillance colonoscopy programme aimed at detecting and treating any signs of early cancer. This review describes the current known risk factors associated with this increased risk, explores our current molecular understanding of cancer development and reviews potential new methods (molecular and technological) designed to help the surveillance programme.

**Abstract:**

Patients with colonic inflammatory bowel disease (IBD) are at an increased risk of developing colorectal cancer (CRC), and are therefore enrolled into a surveillance programme aimed at detecting dysplasia or early cancer. Current surveillance programmes are guided by clinical, endoscopic or histological predictors of colitis-associated CRC (CA-CRC). We have seen great progress in our understanding of these predictors of disease progression, and advances in endoscopic technique and management, along with improved medical care, has been mirrored by the falling incidence of CA-CRC over the last 50 years. However, more could be done to improve our molecular understanding of CA-CRC progression and enable better risk stratification for patients with IBD. This review summarises the known risk factors associated with CA-CRC and explores the molecular landscape that has the potential to complement and optimise the existing IBD surveillance programme.

## 1. Introduction

Inflammatory bowel disease (IBD) is a chronic relapsing-remitting disorder of the gastrointestinal tract, comprising two main subtypes: ulcerative colitis (UC) and Crohn’s disease (CD). The aetiology of IBD is thought to be multifactorial, with the disease arising following incompletely defined environmental triggers in genetically predisposed individuals and resulting in an abnormal immune-driven inflammatory response towards altered gut microbiota [1,2].

Approximately 7 million people are affected with IBD worldwide [3], with a prevalence that surpasses 0.3% [4]. In the UK it is estimated that 300,000 people suffer from IBD [5], whilst the highest reported prevalence values have been seen in Europe (UC 505 per 100,000 in Norway; CD 322 per 100,000 in Germany) and North America (UC 286 per 100,000 in the USA; CD 319 per 100,000 in Canada) [4].

Patients with IBD have an increased risk of developing colorectal cancer (CRC) [6,7]. The first large meta-analysis in 2001 by Eaden et al. [6] assessing CRC risk in patients with IBD showed a risk of 2% at 10 years after UC diagnosis, 8% at 20 years and 18% at 30 years, with an overall CRC prevalence of 3.7% [6]. The largest population study to date (*n* = 96,447), demonstrates a 1.7-fold increased risk of developing CRC with an overall adjusted HR of 1.66 (95% CI 1.57–1.76) and a 1.6-fold increased risk of dying from CRC for patients with UC compared with the general population [8].

However, more recent studies suggest that CRC rates in patients with UC might be declining [9,10,11,12,13]. Specifically, a meta-analysis by Castaño-Milla et al. [9] demonstrated a reducing incidence rate from 4.29/1000 patient years with UC in the studies published in the 1950s to 1.21/1000 patient years in studies published in the last decade, whilst a meta-analysis from Lutgens et al. [12] showed a pooled standardised incidence ratio of CRC in all patients with IBD in population-based studies was 1.7, with cumulative risk of CRC of 1%, 2% and 5% after 10, 20 and >20 years of disease duration, respectively [12]. Both recent studies suggest lower incidence of CRC in IBD patients than that described in 2001 [6]. However, studies from St. Mark’s Hospital in London, a tertiary referral centre, which has the longest running surveillance programme in the world, suggest that although there was a decrease in incidence of advanced CRC and interval CRCs over the past 4 decades, the incidence rate of early CRC and precursor dysplasia has increased 2.5-fold compared to the last decade [14]. Furthermore, a large Scandinavian study examining a cohort of 96,447 patients from 1969–2017, also suggested that individuals with UC are diagnosed with less advanced CRC compared to the control group of non-IBD patients [8], data that was supported by a recent Cochrane systematic review and meta-analysis [15]. The variable quantification of the increased risk of CRC in IBD patients in the aforementioned studies likely reflects differences in country of origin, study population, disease duration [16], more advanced surveillance and treatment techniques and intercountry colectomy rates [8], as well as bias arising from selection, surveillance and lead-time [8].

Despite uncertainty about the precise CRC incidence in IBD, the overall association between IBD and an increased risk of developing CRC is well established and largely accepted in clinical practice [6,7,17], hence IBD patients are enrolled into endoscopic surveillance programmes that aim to detect early signs of cancer and offer treatment to patients in a timely manner.

It has been more than 40 years since the IBD colonoscopy surveillance program was introduced in the UK [18], and the aforementioned studies that report the declining incidence of colitis-associated CRC (CA-CRC) may reflect the advances in endoscopic surveillance across this time. Current UK, European and American endoscopy guidance suggests this should commence at between 6 and 10 years following IBD diagnosis [19,20,21]. Chromoendoscopy (CE) is now the recommended gold standard surveillance technique (over white light endoscopy (WLE)) [14,19] and it has been suggested that the decreased incidence of interval cancers in the past decade is a result of increased use of CE across this same time period [14].

Colonoscopy has played a huge role in surveillance, cancer prevention and early detection [15]. Along with advances in endoscopic technology (high definition, narrow band imaging and CE) and endoscopic management (polypectomy/endoscopic submucosal dissection (ESD)/endoscopic mucosal resection (EMR)), it is also possible that other external factors, such as methodological aspects (hospital-based vs. population-based studies), better medical treatments (use of drugs with chemoprotective effects), better disease control and higher colectomy rates, may have also contributed to the reduction of CA-CRC incidence in the last decade [8,22]. It should be underlined that prospective multicentre randomised controlled trials (RCTs) with long periods of patient follow-up are required to accurately assess the efficacy of new methods in reducing CRC related deaths.

The IBD surveillance programme is fraught with challenges. In IBD, neoplasia is thought to often arise from a field of flat dysplasia with indistinct margins, in an environment of inflammation, scarring and post-inflammatory polyps, making endoscopic detection and resection challenging [23,24], often requiring highly specialist expertise. In practice, histological grading of dysplasia suffers from considerable interobserver variability and it can be challenging to differentiate between regenerating epithelium [25,26], sporadic adenoma and UC-associated dysplasia [27]. Furthermore, the IBD surveillance colonoscopy programme is labour-intensive and costly (approximate cost to the UK NHS is £17,557/quality-adjusted life year [28,29]). A recent national survey of endoscopic practice in the UK highlighted the increased pressure on services [30].

Data from the St. Mark’s IBD surveillance registry demonstrates the 10-year CA-CRC risk of LGD (*n* = 156/1375) and HGD (*n* = 48/1375) at 30% and 50% respectively, and a significantly increased CA-CRC risk compared to those with no neoplasia (1.5%) and/or sporadic adenomas (*n* = 85/1375, 6.5% risk) [14]. However, many CA-CRC patients do not have a preceding dysplasia diagnosis (39%) [14], interval cancers in IBD are reported to be up to 30% (despite adherence to surveillance protocols) [31] and we have also seen that the chance of identifying undetected cancer at colectomy can be up to 25% in LGD and 50% for HGD [14]. Although these numbers are worrying, it has also been reported that most IBD patients do not develop dysplasia and the majority of LGD lesions do not progress to cancer (79.6% survival rate at 10 years) [14] with over 50% showing regression, albeit the mechanism remaining unclear [32,33]. This demonstrates a lack of molecular understanding of disease progression, and highlights the suboptimal methods of current surveillance practice and our inability to accurately predict cancer risk in IBD.

This review will firstly describe our current understanding of the various risk factors associated with neoplasia and CA-CRC development in patients with IBD, before discussing how current clinical practice aims to implement these for risk stratification. Finally, we highlight the potential new methods of assisting and optimising the surveillance programme so we can best predict cancer occurrence.

## 2. Risk Factors for Neoplasia Development and Progression to CA-CRC

The likelihood of cancer forming is predicted by several risk factors acting in combination. Clinical assessment of these in IBD patients partly dictates the frequency of surveillance colonoscopies. Below we describe the prominent risk factors currently considered in clinical decision-making (see also Table 1).

### 2.1. Disease Duration

One of the most important risk factors for CA-CRC is disease duration. The 2001 meta-analysis from Eaden et al. [6] demonstrates a risk of 2% at 10 years, 8% at 20 years and 18% at 30 years after colitis onset. Further meta-analyses and other population cohort studies have supported these findings [17,22,34,35,36,37,38].

As already mentioned, CA-CRC incidence rates have declined over the past four decades, but data still demonstrate a cumulative increase in CA-CRC risk with disease duration. For example, data from the St. Mark’s Hospital surveillance cohort report that the cumulative incidence of CRC was 0.1% in the first decade (since UC symptom onset), followed by 2.9%, 6.7% and 10% by the second, third, and fourth decade, respectively [14]. Results that have been mirrored in the more recent meta-analysis [9,10,12] as mentioned above. Whilst the mechanism underlying this association is not completely understood, it is possible that the increased duration of exposure to repeated cycles of inflammatory insults and epithelial regeneration could be a contributor (as explained further below in Section 2.3 and Section 5.2).

### 2.2. Disease Extent

Increased disease extent in UC has been associated with increased risk of developing CA-CRC. In the landmark study by Ekbom et al. [7], more than 3000 patients with UC were followed up and the risk for CRC increased from 1.7, 2.8 and 14.8-fold in patients with proctitis, left sided colitis and pancolitis, respectively (as compared to the general population). The recent Scandinavian study by Olén et al. [8] also demonstrated an increased risk of CRC incident in patients with more extensive colitis (1.88, 1.72–2.07) compared to those with left sided or proctitis. Most studies including more recent meta-analysis have come to similar conclusions [6,17,34,35], with one study demonstrating a pooled univariable OR of 2.43 (95% CI 2.01–2.93) for extensive disease compared to left sided colitis [39]. As for disease duration, the mechanism underlying the association between disease extent and CA-CRC risk is not completely clear; however, it is possible that it is simply a result of the increased surface area of colonic cells exposed to inflammatory insult, and subsequent increased pervasiveness of epithelial repair and dysbiosis.

A recent Scandinavian study that included 47,035 patients with CD and 463,187 controls demonstrated a HR of 1.40 (95% CI 1.27–1.53) of incident CRC diagnosed in patients with CD [40]. Whilst the heterogenous phenotype of CD makes standardising and comparing the effect of disease extent in this cohort difficult, it has been shown that there is an increased CA-CRC risk when more than 30–50% of colonic mucosa is involved in CD [41], with similar risks if extent and duration are comparable to UC [42,43,44,45]. However, in one study from the early 1990s, extensive CD (defined as involving more than two thirds of the colon, [*n* = 22/24]) was not found to be associated with higher risk of CRC when compared to non-extensive CD (involving less than one third of the colon) (OR 0.35, 95% CI 0.01–11.08) [46]. Along with the heterogenous nature of CD as mentioned above, population size and advances in medical management could account for differences seen between these studies across the decades.

### 2.3. Inflammation and Disease Activity

Inflammation and disease activity are also associated risk factors for neoplasia. Rutter et al. [47] demonstrated increasing endoscopic (OR 2.5 95% CI 1.4–4.4) and/or histologic (OR 5.1 95% CI 2.4–11.1) inflammation scores (as defined by a 1-unit increase in their respective scores) were associated with increased risk of CRC in univariable analysis. Whilst in another study, increase in the mean Nancy histologic index during follow-up review was also associated with CRC development (per 1-unit increase, OR 1.69; 95% CI, 1.29–2.21; *p* < 0.001) [48]. In a retrospective single-centre study, it was shown that the cumulative effect of repeated inflammatory insults (as opposed to the single most recent previous endoscopic assessment), represented as a cumulative inflammatory burden (CIB), is an important risk factor for colorectal neoplasia in UC [49]. This study demonstrated a HR 2.1 (*p* < 0.001) per 10-unit increase in CIB (defined by; the average microscopic inflammatory score between each pair of surveillance episodes multiplied by the surveillance interval in years) and a HR of 2.2 (*p* < 0.001) from a mean severity score calculated from all colonoscopies in the preceding 5 years [49].

The association of increased severity of endoscopic and histological inflammation with CRC risk has also been confirmed in other studies [22,50,51] and supported by meta-analyses [39,52,53]. In recent meta-analysis by Wijnands et al. [39] a pooled univariable OR of 1.98 (95% CI 0.68–5.73) was demonstrated in studies assessing the impact of more severe histological inflammation on CA-CRC risk.

### 2.4. Primary Sclerosing Cholangitis, Stricturing Disease and Family History

Primary sclerosing cholangitis (PSC) is one of the most consistent risk factors reported for CRC in patients with UC [54]. An absolute risk of up to 31% [54], and a 4-fold increased risk (OR 4.79, 95% CI 3.58, 6.41) compared with patients without PSC has been reported [55]. Subsequently, patients with UC and concomitant PSC are advised to undergo annual surveillance [19,54]. Multiple studies support this association [10,22,38,56,57,58,59] with meta-analysis reaching comparable conclusions (OR 4.14, 95% CI 2.85–6.01) [39,55,60,61].

Stricturing disease and post-inflammatory polyps are indicative of previous severe inflammation and are consequently used as surrogate markers for an increased risk of developing dysplasia and CA-CRC [41,62,63,64,65,66]. Recent meta-analysis has demonstrated an OR of 7.78 (95% CI 3.74–16.18) in all pooled data from both UC and CD assessing the association between stricturing disease and CA-CRC [39]. Furthermore, a family history of CRC has been associated with an increased risk of developing CA-CRC [22,67,68,69], with recent meta-analysis illustrating an OR of 2.59–2.62 depending on how studies had defined family history of CRC (studies included a range of definitions for family history including those of only first-degree relatives, second-degree relatives or ‘any relative’ above and below the age of 55) [39]. Patients with such features are reviewed more frequently in the surveillance programme [19,54].

### 2.5. Dysplasia

Data regarding the risk of advanced neoplasia after the discovery of visible LGD are limited and variable, often generated from studies with small cohorts. St. Mark’s registry data (*n* = 1375) reveal that the 10-year cancer risk conferred by detection of LGD and HGD is 30% and 50%, respectively, and with a statistically significant (*p* < 0.001) increased risk compared to those with no neoplasia (2%) and/or sporadic adenomas (7%) [14]. A recent large nationwide cohort with long-term follow-up (*n* = 4284 IBD patients with LGD), demonstrated a cumulative incidence of advanced neoplasia of 21.7% after 15 years [70]. Smaller population cohort studies examining LGD and HGD, over variable follow-up periods, show wide and variable rates of neoplasia progression (ranging between 0–54%) [66,71,72,73,74,75,76].

Earlier meta-analysis from Fumery et al. [60] calculated a pooled CRC rate after LGD of 0.8 per 100 patient-years follow-up. However, most of the studies included had limited numbers of LGD patients (range 2–172) and no studies including >60 patients exceeded a median follow-up duration of 5 years [60]. Another meta-analysis also alluded to the very small numbers of heterogenous studies involved, with subsequent low quality of evidence [77], whilst a most recent meta-analysis, although citing a wide variation in results, showed pooled univariate ORs of 10.7–10.85 (95% CI 4.60–24.87) [39].

There is some evidence to suggest that recurrent dysplasia or recurrently normal endoscopic findings are associated with increased and decreased risk of CRC respectively [62,78,79]. Whilst LGD lesions that are non-polypoid (HR 8.6, *p* < 0.001), endoscopically invisible (HR 4.1, *p* = 0.02), large (≥1 cm) (HR 3.8, *p* = 0.01) or preceded by indefinite dysplasia (HR 2.8, *p* = 0.01), metachronous or multifocal have also been found to be independent risk factors for developing HGD or CRC in patients with UC diagnosed with LGD [60,66,71,80]. Large prospective multi-centre studies are required to validate these findings and more accurately demonstrate the association between dysplasia and progression to CRC [81].

### 2.6. Other Associated Risk Factors

There is some debate as to whether younger age at onset and childhood onset of UC are associated with an increased risk of CRC, irrespective of disease duration. Although studies show conflicting results [22,41,59], early studies suggest a trend towards this increased risk [6,7], which have been supported by more recent retrospective population data [82] including meta-analysis confirming these results by demonstrating standardised incidence ratios (SIRs) of 8.6 (95% CI, 3.8–19.50) [17]. A large Scandinavian study by Olen et al. [8] also demonstrated an especially high HR of incident CRC for childhood-onset UC (HR 37.0, 95% CI 25.1–54.4), compared to patients with elderly-onset UC (0.98, 0.88–1.08); however, it is unclear whether these groups had comparable disease duration.

Certain medical therapies have been shown to be protective against neoplasia progression with 5-aminosalycilates (5-ASAs) being identified as potential chemo-preventative agents. Although population studies show conflicting results [47,83,84,85,86], multiple meta-analyses support the use of 5-ASAs as a chemo-protective agent in IBD and reach similar conclusions (OR 0.58, 95% CI: 0.45–0.75) (OR 0.63; 95% CI: 0.48–0.84) [39,87,88,89,90]. Although earlier studies show limited chemo-preventative effect with immunomodulators [91,92,93], there has been increasing evidence to support the potential chemo-preventative effect of thiopurines [94,95], with more recent meta-analysis reaching the same conclusions and demonstrating an OR of 0.67 (95% CI 0.45–0.98) [96] and 0.49 (95% CI 0.34–070) [97], respectively. Six studies have evaluated the chemo-preventative potential of TNF-alpha inhibitors, with meta-analysis not demonstrating any protective effect [39]. It is overall unclear whether reaching the goal of endoscopic/histologic healing is the important factor responsible for decreasing the risk of CA-CRC, or whether it is the application of different drugs/drug classes and their specific effects. Multiple reviews advocate the use of 5-ASAs as chemo-preventatives in UC, but data on thiopurines or biologics is somewhat limited, and in general there is a lack of RCTs for all medications in this context [98,99].

Finally, meta-analysis has also revealed that geography could also be a significant risk factor with higher rates of CA-CRC seen in the USA, UK and Asia compared with Scandinavia and other countries [6,35,37]. Whether this is due to the expansion of the ‘typical westernised diet and lifestyle’ into Asia, is still to be determined. From an ethnicity standpoint, there appears to be no difference in CRC incidence between Caucasian and African-American IBD patients [100], with more work needed to understand how different patient demographics play a role in disease incidence and progression.

**Table 1 cancers-13-02908-t001:** Summary of Clinical risk factors associated with increased risk of CRC in patients with UC and CD.

Risk Factor	Odds Ratio	Reference
Disease duration		
Risk increases with duration of disease, most apparent after 6–8 years, with a cumulative stacked effect.	4.74	[6,7,39]
**Disease extent**		
UC: Increasing risk respectively from pancolitis > Left sided colitis > proctitis	2.43	[7,39]
CD: Increased risk has been demonstrated with more extensive disease	N/A	[39,40,41]
**Inflammation and severity**		
Risk increases with increasing disease severity (endoscopic). Even more apparent with cumulative inflammatory burden.	2.62	[39,47,49]
Risk increases with increasing disease severity (histological).	1.98	[39,47]
**Primary Sclerosing Cholangitis**		
Associated with increased risk requiring annual surveillance from diagnosis.	4.14	[39,54]
**Family history**		
Associated with increased risk depending on age of diagnosis and degree of relative.	2.62	[39,67,68]
**Stricturing disease and post inflammatory polyps**		
Surrogates for previous severe inflammation and associated with higher risk	7.78 and 3.29	[39,62,63,101]
**Dysplasia**		
Associated with increased and variable risk, with pooled CRC rate after LGD of 0.8 per 100 patient-years follow-up.	10.7	[14,39,60]
**Age of onset and sex**		
Earlier age of onset (<16 yoa) is associated with increased risk. Males have a slightly higher risk.	1.27	[6,7,8,17,39,82]

N/A—Not available.

## 3. Current Practice in IBD Assessment and Surveillance

### 3.1. Clinical and Biochemical Assessment

As described above, there are many potential risk factors for CA-CRC development in IBD. These factors, along with a clinical, biochemical and endoscopic/histopathological assessment, are used to estimate a patient’s CRC risk and surveillance intervals are tailored accordingly [19,21,54] (Table 2). The Montreal classification for UC describes the extent of disease. There are additionally disease activity indices, such as the Mayo score and Truelove and Witts Severity Index, which are used to assess a patient’s clinical status objectively [19,21,54].

C-reactive protein (CRP), erythrocyte sedimentary rate (ESR), albumin and a full blood count (haemoglobin, white cell count and platelets) are established serum biomarkers that indicate the level of systemic inflammation present, and in the case of IBD act as surrogate markers of disease activity. Studies on the relationship between these serum inflammatory biomarkers and the development of CA-CRC in UC have shown significant correlations. For example, a study by Koutroubakis et al. [102] demonstrated through multivariate logistic regression analysis how patients with CA-CRC (*n* = 55/773) had an increased CRP-albumin score compared to those without CRC (OR 2.40 (95% CI: 1.34–4.30)). This study also demonstrated that this group also had higher median levels of CRP and ESR, and lower haemoglobin and albumin levels, compared to those without CRC [102]. Another study by Ananthakrishnan et al. [103] showed similar results by demonstrating a significant increase risk of CRC across quartiles of CRP and ESR elevation (OR 2.72, 95% CI 0.95–7.76, *p* = 0.17, and OR 2.06; 95% CI, 1.14–3.74, *p* = 0.007). Interestingly, CRP consistently and significantly increases with progression of stage in sporadic (non-IBD) CRCs [104], suggesting it may be a universal indicator of CRC progression. Although these studies suggest an association between higher levels of inflammatory markers and the presence of more advanced disease, their use as a predictive biomarker is still to be determined. RCTs with long-term follow up are required to demonstrate the predictive value of these parameters for CA-CRC.

Faecal calprotectin (FC) is another biomarker of inflammation used in clinical practice for the diagnosis and monitoring of IBD activity [105,106,107,108]. However, applying FC as a biomarker for diagnosing or detecting CRC has proved unreliable. One study demonstrated that low FC accurately identified IBD patients without colonic inflammation in whom CRC surveillance is most effective [109]. Meta-analysis suggests that FC has good utility in diagnosing IBD; however, it was not recommended as a predictive biomarker for CA-CRC [110].

### 3.2. Endoscopic Assessment

As mentioned above, surveillance intervals are adjusted according to the perceived risk of CRC. Current European guidelines are outlined in Table 2.

Chromoendoscopy (CE) is the recommended gold standard surveillance technique [14,19], with a large retrospective study demonstrating significantly improved neoplasia detection rates that are 2-fold higher (8.4% vs. 4%, *p* < 0.001) in the CE group (*n* = 92/1098) when compared with the white-light colonoscopy group (*n* = 175/4373) [14]. Furthermore, an earlier RCT by Kiesslich et al. [111] showed that CE has a significantly better correlation between the endoscopic assessment of degree (*p* = 0.0002) and extent (89% vs. 52% *p* < 0.0001) of colonic inflammation and the histopathologic findings compared with the conventional white light colonoscopy group. Furthermore, significantly more neoplasia was detected in the CE group (32/84 vs. 10/81 *p* = 0.003) [111]. Other studies have also shown an improved neoplasia and dysplasia detection rate with use of CE [112,113], conclusions that have been supported by meta-analyses [114,115].

It has been suggested that the decreasing rates of interval cancers in the past decade could be attributed to the increased use of CE across this same time period [14]. However, although there is compelling evidence to support CE over white-light endoscopy (WLE), the converse has also been reported, with one retrospective study by Mooiweer et al. [116] demonstrating no significant difference in neoplasia detection rates between CE and WLE groups across 3 tertiary referral centres. Moreover, it must be reiterated that the improved disease control with advances in therapeutics, increased surgery, and management strategies, may also have contributed to the reduction in CRC rates over the last decade.

### 3.3. Histopathological Assessment

Inter-observer variability in histopathological grading of dysplasia in IBD is a well-recognised area of contention [25,26,27], and the grading methods used by histopathologists are not universal, further compounding this issue [117].

Historically, the Geboes Score and the modified Riley Score were commonly used to evaluate the histological disease activity in patients with UC [118], by assessing immune infiltrate, mucosal structure and crypt integrity and reviewing the presence of ulceration/erosions to subsequently classify histological severity of disease [119]. However, newer scoring indexes, such as the Nancy index, is proving to be a potentially prominent and established scoring method to help standardise reporting [120]. This scoring method reviews the presence or absence of ulceration, and the extent of the acute and chronic inflammatory infiltrate [121]. One study by Kirchgesner et al. [48] demonstrated that the mean Nancy index scores could be significantly associated with future development of CA-CRC (per 1-unit increase in the mean Nancy histologic index, OR 1.69; 95% CI 1.29–2.21; *p* < 0.001).

There is evidence supporting that overexpression of p53 in colonic epithelia can delineate colitis-associated (p53+) from sporadic-type dysplasia (p53-) and discriminate between regenerative changes and intraepithelial dysplasia [122]. Furthermore, its application as a tissue biomarker to predict the risk of evolution toward malignancy and detecting early-stage neoplasia has also been shown [122,123,124,125]. Meta-analysis has shown that immunohistochemical p53 expression in both patients with UC and dysplasia (compared to patients with UC and without dysplasia; *n* = 297 vs. 335, OR 10.76, 95% CI 4.63–25.03, *p* < 0.001) and no dysplasia (compared to normal tissue; *n* = 233 vs. 130, OR 3.14, 95% CI 1.58–6.24, *p* = 0.001) was closely associated with CA-CRC progression [126].

Other studies have demonstrated that expression of p53 in combination with other markers (p21, AMACR, Ki67) can increase the value of p53 as a tissue biomarker for dysplasia diagnosis and CA-CRC progression [122,127,128]. One study questioned the feasibility and practicality of utilising p53 as an effective biomarker by suggesting that neither immunohistochemistry (IHC) or sequencing have the full capability to predict p53 status alone for patients with CRC, but a combination of these two technologies is required to provide a more complete assessment of p53 status [129]. This highlights that although immunostaining of p53 may be a valuable diagnostic tool for detecting dysplasia in UC, it must be noted that additional molecular methods may be needed to accurately assess p53 status in colitis-associated dysplasia [130].

Furthermore, its utility in detecting CA-CRC has also been questioned with patients on concomitant immunosuppressive therapy [131], with some reports suggesting that such treatment might result in low antibody response rates due to their interaction with multiple steps of the inflammatory cytokine cascade [131,132,133,134]. Further studies are required to validate p53 as a biomarker for neoplasia, whether that be in isolation, or combined with other molecular tests as above.

## 4. Molecular Understanding of Progression to CA-CRC

The adult colon can range between 100–150 cm in length with a circumference of 6–10 cm [135]. It is lined by approximately 10 million crypts, which are single cell layer invaginations composed of 1000–4000 columnar epithelial cells [136,137]. At the base of each crypt there are estimated to be around five to seven rapidly cycling LGR5-positive stem cells [136,138,139]. Routine biopsies at surveillance colonoscopy are taken in a targeted fashion or quadrantic biopsies are taken every 10 cm along the length of the colon, and therefore will only sample a very small percentage of the colon (approximately 0.001%). Thus, pre-neoplastic changes may be missed, a challenge that is further compounded by the technical difficulties of a colonoscopy, the inter-observer variability in histopathological interpretation and the lack of molecular understanding of disease progression, as mentioned above. Molecular medicine, novel biomarkers and specific mucosal profiling could be valuable tools for improving risk prediction in these patients (Figure 1).

### 4.1. Field Cancerisation

Field cancerisation was first described over 50 years ago [140] and is defined as the process by which the normal cell population is ‘replaced by a cancer-primed cell population that may show no morphological change’ [141]. Field cancerisation is recognised as an established mechanism underlying the development of CA-CRC [141,142], and has been demonstrated in the premalignant non-dysplastic colon [136,143,144], occurring as either a single large clonal expansion or via the parallel expansion of multiple large clones (clonal mosaicism) [32,143].

Leedham et al. [143] observed that the genomic alterations in individual crypts within different neoplastic lesions of multifocal dysplasia showed a common clonal origin. Furthermore, they demonstrated that some adjacent (phenotypically normal) crypts also shared the same driver mutations [143], suggesting a field of ‘cancerised’ cells that was macroscopically and histopathologically normal. Galandiuk et al. [144] demonstrated how a ‘cancer primed’ phenotypically non-malignant region can progress to CA-CRC by describing one case where at least 3 separate *TP53* mutations arose in non-neoplastic colon, each of which gave rise to neoplastic lesions several years later, with the most recently detected *TP53* mutant spreading over 3 years to involve the entire length of the colon. A study by Salk et al. [145,146] provided further evidence for a cancer-primed field by using mutations of hypermutable polyguanine repeats as markers of clonal populations, to demonstrate patients with HGD or CRC had evidence of larger pre-cancer clonal expansions across the colon. Again, some of the clonal patches from wherein a cancer arose involved a large surface area of the colon [136,145,146]. Lai et al. [147] used array comparative genomic hybridisation (aCGH) to show that chromosomal instability in phenotypically normal mucosa increases with proximity to dysplasia and cancer [23]. There have also been reports of early aneuploidy field changes in the non-dysplastic UC epithelium occurring in the genesis of CA-CRC [147,148,149,150,151,152]. Remarkably, these aneuploid field changes can occur without morphological or histological change, highlighting that histopathology of tissue alone is likely an inadequate biomarker of cancer risk as it misses what are presumably “cancer-primed” populations of cells [34].

Kakiuchi et al. [153] have demonstrated the possibility that the inflamed intestine, under chronic relapsing-remitting inflammation, undergoing repeated destruction and regeneration, can evoke a tissue response by eliciting the remodelling of tissues through positive selection of clones that acquire mutations. By performing whole exome sequencing (WES), mutations that commonly involve the *NFKBIZ*, *TRAF3IP2*, *ZC3H12A*, *PIGR* and *HNRNPF* genes and implicated in the downregulation of IL-17 and other pro-inflammatory signals were identified and shown to expand across the entire rectum [153]. Areas of up to 19 cm^2^ across the rectum were completely reconstituted by a few clones that carry IL-17 pathway-related mutations in a field cancerisation manner [153]. Nanki et al. [154] were able to corroborate this study by using WES of 76 clonal human colon organoids to identify a unique pattern of somatic mutagenesis in the inflamed epithelium of patients with UC. This study also demonstrated the pervasive acquisition and spread of mutations, related to IL-17 signalling, that are presumed to facilitate selective expansion of these cells under inflammatory conditions [154].

With the concept of field cancerisation in mind, we are beginning to appreciate that a polypectomy or limited resection (ESD/EMR) is unlikely to be totally effective in reducing cancer risk in IBD as they have the potential to leave behind a field of cancer-primed tissue not seen macroscopically or histologically, rendering a colectomy the only long-term “cure”. However, measurements of the cancerised field could become a new trackable biomarker to allow for optimised risk stratification, individualised case-by-case management and clinical decision making [141,155].

### 4.2. DNA: Gene Panels and Copy Number Alterations

Whilst the phenotypic differences between CA-CRC and S-CRC have been demarcated [7,10,23,24,156,157], and the basic biology of S-CRC well studied [158,159,160,161], the molecular differences between CA-CRC and S-CRC are not completely understood. It is particularly important to understand these differences from the perspective of clinical management.

As previously mentioned, aneuploidy is a candidate for a molecular biomarker that has been associated with field cancerisation and CA-CRC. Mutational panels of key driver genes such as *TP53*, *APC*, *KRAS*, *CDKN2A* and *FBXW7* [136,144,162,163] have also been suggested, but whilst a mutation in one of these genes may not be able to differentiate a S-CRC from CA-CRC, a mutation of one of the above found from dysplastic tissue in a patient with UC still might be able to predict CA-CRC occurrence [164,165]. The *TP53* mutation is typically an early event in CA-CRC [162], whilst *APC* [164] and *KRAS* [165] mutations are reported to be less prevalent in CA-CRC compared to S-CRC. So far, only aneuploidy, as measured using flow cytometry, has been shown to carry predictive potential of CRC in IBD [148,150,166,167]. Meta-analysis has also revealed that aneuploidy is an effective parameter for individual cancer risk (OR 5.17, 95% CI 2.28–11.71, *p* < 0.0001) [39], whilst other meta-analysis illustrates that the combined assessment of dysplasia and aneuploidy outperforms the use of each parameter alone (OR 8.99, 95% CI 3.08–6.26) [149].

A study by Baker et al. [23] used multi-region sequencing to demonstrate the level of genetic diversity in IBD neoplasias (both in terms of inter-tumour variation of genomic alterations and intra-tumour genomic heterogeneity in established cancers [136]) and described that CA-CRCs are molecularly distinct from their sporadic counterparts. Specifically, they reported that CA-CRCs have a slightly increased burden of single nucleotide alterations (SNAs) compared with S-CRCs (though this was not statistically significant in their cohort), but importantly these include recurrent mutations in genes infrequently mutated in S-CRCs [23]. This has been supported by meta-analysis [168] and other studies that revealed a distinct set of genes bearing SNAs in CA-CRCs [169,170,171,172]. Several CA-CRC-specific gene mutations, including SOX9, EP300, NRG1, and IL16 have been identified, supporting the notion that CA-CRC has a unique genetic composition compared to S-CRC [169,173].

Ulcerative colitis can be thought of as an aging disease of the colon, compounded by the repeated cycles of inflammatory insult and epithelial regeneration [136,173]. Quantitative fluorescence in situ hybridisation (qFISH) and quantitative polymerase chain reaction (qPCR) of colonic epithelium from patients with UC has been used to demonstrate accelerated telomere shortening compared to leukocyte or adjacent stromal controls [173,174,175]. Telomere shortening results in chromosomal instability [176]; however, the techniques and results from the above lack either the high specificity demanded by a CRC biomarker (qPCR) (refer also to Section 4.5) or the capability for high throughput (qFISH) to create a clinically applicable test [173]. On the other hand, utilising shallow whole genome sequencing (sWGS) can provide high resolution genome-wide copy number alteration (CNA) profiles [23]. This has been shown to be cost effective and capable of high throughput analysis [23,177,178], thus providing a potential tool to assess CNA burden and potentially aneuploidy as a biomarker in IBD.

Moreover, CA-CRCs appear to have a high burden of CNAs with recurrent losses and gains that are also distinct from S-CRCs [23]. This has been demonstrated at arm level, where six chromosomal arms were significantly more likely to be gained in CA-CRC compared with S-CRC and 10 arms were significantly more likely to be lost, most notably 5q (57% vs. 17%; OR 5.87; *q* < 0.001) and 17q (37% vs. 15%; OR 3.34; *q* = 0.01) [23]. This was also shown in a study that examined chromosomal instability using aCGH and found increased numbers of CNAs was associated with progression to CA-CRC [179].

### 4.3. RNA: Transcriptomics

RNA expression is variable, highly modulated and confounded by cell type, making its utility as a molecular biomarker potentially limited. However, using RNA analysis to map the phenotypic landscape along high-risk IBD colons has the potential to reveal the mechanisms underlying disease progression and improve our understanding of the biology of CA-CRC. In one study, microarrays have been used to investigate gene expression changes in UC neoplastic progression by comparing the mucosa of non-dysplastic, dysplastic and cancerous colonic tissues [180]. Five genes were found to be common to UC dysplasia and CA-CRC relative to non-dysplastic UC (*CCND1*, *SERPINB6*, *PAP*, *IL8* and *NOS2A*) [180].

Similarly, a study by Pekow et al. [181] demonstrated that nine genes (*ACSL1*, *BIRC3*, *CLC*, *CREM*, *ELTD1*, *FGG*, *S100A9*, *THBD* and *TPD52L1*) were progressively and significantly up-regulated from controls (*n* = 11) compared to non-dysplastic (*n* = 4) and neoplastic (*n* = 11) patients with UC. Furthermore, this study found 468 genes were significantly up-regulated and 541 genes were significantly downregulated in patients with UC and neoplasia compared to patients with UC and without neoplasia [181]. Immunostaining was also used to validate microarray analysis and found increased expression of *S100A9* and *REG1α* in CA-CRC and in non-dysplastic tissue from patients with UC harbouring remote neoplasia, compared to patients with UC and without neoplasia and controls [181]. This study therefore demonstrates field cancerisation, and suggests that detection of this field using expression analysis could also be used as a tool to identify patients concealing CA-CRC. Watanabe et al. [182] also identified 20 genes showing differential expression in CA-CRC (*n* = 10) and UC control patients (*n* = 43), which included cancer-related genes such as *CYP27B1*, *RUNX3*, *SAMSN1*, *EDIL3*, *NOL3*, *CXCL9*, *ITGB2* and *LYN*. Specific roles for *TIMP1*, *ZEB1* and *CD133* expression have also been proposed in the development of progressive disease [183,184,185]. Whilst these studies suggest differences in gene expression between normal and neoplastic tissue and provide further insight into the potential mechanistics involved with disease progression, the heterogeneity in their results must be noted. Their differences could be explained by the relatively small population sizes of their neoplastic groups, whilst it should also be remembered that variation in patient group demographics, disease activity, sample collection and processing may affect RNA expression from sample to sample, therefore making direct comparisons difficult.

Large RCTs are required to validate these findings, and to improve our molecular understanding of transcriptomic dynamics in disease progression and chemoprotective properties of potential therapeutic strategies. These studies suggest that it may be possible to incorporate gene expression analysis in the identification of patients with UC at high risk of progressive disease, with subsequent implications of improving the efficacy of surveillance and management of these patients [182].

### 4.4. Methylation

Chronic inflammation has been demonstrated to promote aberrant DNA methylation in conditions such as UC [186,187]. Furthermore, it has been shown that a clone with altered methylation status can undergo clonal expansion [141,188,189,190].

A multicentre study by Beggs et al. [191] has demonstrated that the detection of DNA methylation using a five-marker panel (*SFRP2*, *SFRP4*, *WIF1*, *APC1A*, *APC2*) was accurate in detecting pre-cancerous and invasive neoplasia (AUC = 0.83; 95% CI: 0.79, 0.88), and dysplasia (AUC = 0.88; 95% CI: 0.84, 0.91). Moreover, this study also demonstrates epigenetic field cancerisation in non-neoplastic mucosa (by using a four-marker panel *APC1A, SFRP4, SFRP5, SOX7*) with modest accuracy (AUC = 0.68; 95% CI: 0.62, 0.73) in predicting associated bowel neoplasia through the methylation signature of distant non-neoplastic colonic mucosa [191]. Furthermore, other studies have shown that hypermethylation of *ER*, *MYOD*, *CDKN2A* (encoding p16) and *BVES* could also be detected in the non-dysplastic tissues from patients with UC with progressive neoplasia compared to non-progressors [192,193], providing further evidence of an epigenetic field cancerisation effect.

A systematic review supports these findings, by suggesting that certain individual genes are highly methylated in CA-CRC; RUNX3, MINT1, MYOD, CDKN2A exon1 and the promoter regions of EYA4 and ESR [194]. Whilst other studies suggest ITGA4 and TFPI2 [195], APC, CDH13, MGMT, MLH1 and RUNX3 [196], SLIT2 and TMEFF2 (*p* = 0.05 and *p* = 0.03, respectively) [197], FOXE1 and SYNE1 [198] could also be methylation biomarkers for CA-CRC. Studies from Toiyama et al. [186,199] corroborate these findings by demonstrating how both DNA and RNA methylation profiles could potentially be useful as CA-CRC biomarkers.

Other epigenetic changes, including histone modifications, chromatin remodelling, and non-coding RNAs, are potentially significantly involved in CA-CRC development and progression [200], with more studies needed to verify and validate these findings.

### 4.5. Clonal Evolution and Evolutionary Dynamics

As we have described above, there is great potential for the development of novel molecular biomarkers for neoplasia progression in IBD. However, the ‘extensive within-lesion genetic and phenotypic heterogeneity that are now considered hallmarks of both carcinogenesis [201] and pre-malignant pathology’ [32,202] and the inter-tumour variation of genomic alterations seen in IBD-CRCs [136] are likely to confound an approach based around detection of a single molecular change. Instead, measuring the evolutionary dynamics of pre-cancer clones offers a potentially more robust route to efficacious biomarkers of cancer risk.

One example of a candidate ‘evolutionary biomarker’ is genetic or clonal diversity [32]. The hypothesis is that more genetic diversity across an inflamed colon is likely to correlate with a greater probability of the colon containing a clone that has evolved closer to a cancerous phenotype. To date, no studies have used genetic diversity measures for CA-CRC risk, but the principle has previously been applied to Barrett’s oesophagus, whereby multicolour fluorescence in situ hybridisation (FISH) data was used to assess the genetic diversity at single-cell resolution in non-dysplastic Barrett’s oesophagus patients [32,203]. Previous studies of the clonal composition of individual Barrett’s lesions suggested genetic mosaicism as potential predictors of progressive disease [204,205]. In a recent study by Martinez et al. [203] variance in clonal composition between multiple sample points taken from a single oesophagus at a single gastroscopy, as well as across time by comparing clonal composition between tissue samples taken at two different time points were measured. A minimum of 50 cells per sample were scored for abnormalities by FISH at seven markers including CEP7, CEP17, p53, p16, Her-2/neu, 20q and MYC, and a ‘clone’ was defined as the collection of cells with identical genotype (for example, identical copy-number profiles) [203]. Genetic diversity using ecological diversity measures (Shannon and Simpson diversity indices) assessing the number of different clones present and their abundance was quantified. In this study, higher levels of genetic diversity (clonal diversity) were found to be predictors of cancer risk even after controlling for age, Barrett segment length, *TP53* mutation and aneuploidy status [203,206].

A second approach that also reflects evolvability would be detection of large clonal expansions and field cancerisation as a measure of ongoing evolution [32,136]. In a study by Salk et al. [145] by using 4–5 biopsies (spaced approximately 20 cm apart) to detect mutant clones with shared point mutations it was demonstrated that UC progressors (to HGD or CRC) have substantially larger clonal expansions than non-progressors [32]. Thirdly, changes over time in the clonal composition of the epithelium, irrespective of what those changes might be, may be an indicator of rapid ongoing dynamic evolution [32,136]

The Baker et al. [23] and Beggs et al. [192] studies have revealed that a high SNA burden, CNAs and methylation patterns accrue prior to cancer formation respectively, in non-dysplastic tissue or those with LGD. In the study by Baker et al. [23] it was also demonstrated that known driver mutations were almost exclusively present in the founder lineage of the tumour and that the majority of CNAs occur prior to the onset of cancer growth. This was corroborated by another study demonstrating that chromosomal alterations occur early in CA-CRC, preceding the histological development of dysplasia, and further revealed a relative loss of 18q in progression of UC-associated neoplasia [207]. These findings indicate evolutionary dynamics consistent with the ‘Big Bang’ model proposed for S-CRCs [160,208], where a cancer is formed with all the driver mutations, rather than acquiring them sequentially after the initiation of cancer growth [23,208]. As we have already seen, aneuploidy is well known to precede or co-occur with dysplasia in CA-CRC [148,150,167], and this data along with the numerous examples in the aforementioned studies reporting extensive genetic and epigenomic alterations occurring in morphologically normal intestinal mucosa [23,143,152,191], indicate that clonal evolution of cancer begins before the development of a true and clinically detectable malignancy in IBD [32].

Clearly more studies, involving large multi-centre cohorts are needed to fully characterise the evolution of CNAs, SNAs and methylation profiles in progressor and non-progressor IBD patients. The long-term aim would be to derive a sensitive and specific molecular test, guided by our evolutionary insight, to compliment the already established clinical, biochemical, endoscopic and histopathological predictors of disease progression.

## 5. New Directions

### 5.1. Non-Invasive Tests

#### 5.1.1. Circulating Tumour DNA (ctDNA) and Cell-Free DNA (cfDNA)

Analysis of DNA fragments in blood samples—so called ‘liquid biopsies’—may provide information on disease activity and could *simultaneously* enable the detection and analysis of tumour components without the need for a tissue biopsy [209].

So far, their application to IBD has been limited to one study that has shown utility in mouse models and humans (*n* = 123) [210]. This study demonstrates that plasma cfDNA concentrations significantly correlate with other inflammatory markers, such as the percentage of neutrophils (*p* = 0.0079) and CRP (*p* = 0.0052), as well as being positively correlated with the clinical severity of UC [210].

So far no studies have tested the correlation between cfDNA and dysplasia or CA-CRC, but meta-analysis suggests their potential use in S-CRC as a tool to detect early disease, identify residual disease, assess treatment response, map prognosis, and identify resistance mechanisms [211]. Some studies have shown the potential of cfDNA to identify specific genetic mutations known to be associated with response to targeted therapy in S-CRC, such as detection of *KRAS* mutations [212,213]. Further meta-analysis has demonstrated clear prognostic value of cfDNA measurements, showing overall increased survival in patients with the lowest levels of baseline cfDNA (*p* < 0.0001) [214]. However, other studies have not demonstrated acceptable sensitivity and specificity of utilising cfDNA as a means for the accurate diagnosis of CRC [215,216].

The potential of using cfDNA as serum biomarker for neoplasia is very attractive; however, the studies included in the meta-analysis [211,214] show marked variation in study size and design. A recent systematic review by Cree et al. [217] reviewing cfDNA as a blood-based biomarker cites the largest study to date included 640 patients (of variable cancer types), with a median study size of 65 cases and 35 controls, whilst the bulk of studies (71%) included less than 100 patients. Standardised methodologies for sample collection and processing, including the specifics for blood collection, processing times and techniques, storage conditions, DNA extraction, quantification, PCR and validation in large prospective clinical studies will be required in future work [218]. Moreover, a better understanding of the origin and biology of cfDNA and ctDNA including the impact of apoptosis, necrosis and the active release, should also be explored, to aid in interpreting the translational value of the results [219,220].

Clearly, even in S-CRC, multicentre prospective studies with larger cohorts are required to validate these findings; whereby the correlation of cfDNA levels to the already established serum inflammatory markers (CRP, ESR, neutrophil counts) and to disease activity (using more commonly used scores such as Mayo, UCEIS, Nancy) already currently used in clinical practice can be measured. Interestingly in the case of CA-CRC, the application of a cfDNA biomarker could potentially prove doubly effective in that they could also be used as a surrogate for dysplasia and CRC progression, as well as characterising the severity of inflammation biochemically.

If shown to be effective, and with the advantage of cheap and easy testing protocols, cfDNA could then potentially be used longitudinally to measure the evolutionary dynamics of IBD neoplasia without the need for repeated invasive biopsies. These methods could then be applied when patients are in remission, flare and/or treatment escalation/de-escalation [221,222], and significantly aid in the risk stratification and decision-making process with regard to surveillance and management.

#### 5.1.2. Stool DNA (sDNA)

Another potentially attractive non-invasive biomarker is stool DNA (sDNA). It is estimated that approximately one half of stool is comprised of gut microbiota and that upwards of one million colonic epithelial cells can be isolated from 1 g of stool, which would allow for the unbiased sampling of the genomics and proteomics of a patient’s colonic epithelium, as well as their gut microbiome [173].

Although targeted mutational analysis of sDNA (*TP53*, *APC*, *KRAS*, *BRAF* or *PIK3CA*) failed to discriminate CA-CRC samples from cancer-free IBD (*n* = 50), the methylation of four genes (vimentin, *EYA4*, *BMP3*, *NDRG4*), distinguished patients with UC; with or without neoplasia, suggesting both high specificity and sensitivity for dysplasia detection [223]. A further study showed that methylation of *BMP3* and *VAV3*, relative to *ZDHHC1* methylation could identify patients with CRC and HGD [224,225]. These studies highlight the potential utility of sDNA analysis in assisting in the diagnosis and monitoring of progressive IBD lesions, with the added benefit of cost-effectiveness in combination with chromoendoscopy [226], it would represent a very attractive non-invasive biomarker for tracking neoplasia progression.

In essence the possibility of non-invasive methods of surveillance is a very attractive area of research. By utilising new potential molecular biomarkers such as serum cfDNA and sDNA to complement the already established markers for disease activity (CRP, FC, colonoscopy and histopathology), clinicians could have confidence to safely extend surveillance intervals based on individual circumstance. Moreover, taking advantage of the already routine blood and stool sampling would be an ideal opportunity to acquire this data. It is an exciting area for future research.

### 5.2. Immune Micro-Environment

As already mentioned, extent and severity of inflammation in IBD is strongly predictive of cancer risk [47,49], but the underlying mechanisms whereby inflammation causes cancer evolution have not been fully determined. Recent work suggests inflammation is not directly mutagenic [23], and so instead the carcinogenic effect is likely to be a consequence of selection for hardy and/or rapidly growing phenotypes—these traits are frequently found in cancer cells. Furthermore, immunosuppressant medications are the mainstay of treatment for IBD, used to both induce and maintain remission, but the fundamental dynamics governing the co-evolution of the immune compartment and neoplastic cells during progression are still not completely understood.

In a study by Zhang et al. [227] transcriptomes of 11,138 single T-cells from 12 patients with sporadic colorectal cancer were obtained, and by using single T-cell analysis from RNA sequencing and T-cell receptor (TCR) tracking indices, they were able to demonstrate the various functional, migratory and developmental connections among different T-cell subsets in CRC and showed how T-cell populations may track with cancer clones. This suggests that monitoring of T-cells could be used to infer the dynamics of epithelial clones in IBD, and provide a further means to track and map neoplasia and field cancerisation.

Recent work also suggests an essential role for immune evasion in the transition to malignancy in S-CRC [228,229], whilst Galon et al. [230] have also demonstrated that an ‘Immunoscore’ that quantifies the CD3+ and CD8+ immune infiltrate of a CRC held more prognostic significance than the currently used AJCC/UICC TNM classification. It follows that the specific type of inflammatory infiltrate could be prognostic in IBD, although this is yet to be tested.

As mentioned above field cancerisation has been shown to underpin the progression to CA-CRC, and it has been suggested that aberrant alterations of stromal cells can promote field cancerisation [136,141]. Severity and extent of inflammation are associated with cancer risk in IBD [39,47], and the aberrant changes in the stroma and microenvironment enable field cancerisation by altering the fitness landscape of the epithelial cells [231], and so could provide selective pressures for phenotype adaptation, promoting the clonal expansion of cancerised lineages [141]. Moreover, whilst the microenvironmental differences between normal and dysplastic stromal tissue in UC has been documented [32,232], the immune co-evolution in CA-CRC remains poorly understood [136], and understanding and mapping this transition could also provide a route to prognostic benefit in IBD.

### 5.3. Microbiome

Studies suggest that the microbiome could also be linked to cancer risk and progression of neoplasia in UC [136]. Early evidence from epigenetic studies has demonstrated that Fusobacterium colonisation is associated with pro-carcinogenic methylation in non-neoplastic UC mucosa (OR 16.18, 95% CI 1.94–135.2) [233], a finding that is supported by another study by Gevers et al. [234]. Yang et al. [235] cites *Alistipes finegoldii*, *Atopobium parvalum*, *Peptostreptococcus anaerobius* as novel microbial drivers of colitis and carcinogenesis in murine and human models, whilst Richard et al. [236] demonstrated that in comparison to S-CRC, CA-CRC was characterised by an increase of *Enterobacteriacae* family and *Sphingomonas* genus and a decrease of Fusobacterium and *Ruminococcus* genus.

There are a number of similarities between the microbiome changes seen in S-CRC and CA-CRC [136]. For example, dysbiosis and the reduction in butyrate-producing Firmicutes such as *Faecalibacterium prausnitzii* [136,237,238,239], as well as the increased mucosal abundance of Enterobacter faecalis and *Escherischia coli*, in particular adherent-invasive *E. coli* [136,240,241,242].

These studies indicate that whilst dysbiosis may be associated with cancer risk, it remains unclear whether these microbiome changes represent primary and independent drivers of inflammation and carcinogenesis [136] or are in fact a secondary consequence of the damage and disruption of the epithelium caused by inflammation or neoplasia [136,243]. Furthermore, these studies suggest that whilst there maybe parallels between the microbiome changes in S-CRC and CA-CRC, the microbiome seen in CA-CRC can be significantly different to that of S-CRC or in normal controls [236], suggesting that a particular set or pattern of dysbiosis may not be the appropriate means to measure cancer risk in general, and perhaps promoting a broader evolution-based analysis of the microbiome on an individual case-by-case basis may be more appropriate. Regardless, the gut microbiome remains an important area of research and further multi-centre prospective studies will be required to validate the findings described above.

### 5.4. Artificial Intelligence

Recent studies have shown great promise in large-scale machine learning approaches (Artificial Intelligence (AI)) for computer-aided diagnosis in endoscopy [244]. Already there has been application in a multitude of areas including colonic polyp/lesion detection [245,246,247,248,249,250], and recognising Barrett’s oesophagus, oesophageal squamous cell carcinoma and gastric cancer, as well as application in histological and optical pathology [245].

AI has also been applied to assisting in the diagnosis and severity of IBD, by identifying endoscopic inflammation and grading severity in patients with UC [251,252]. Abadir et al. [245,253] demonstrated that it is possible to create a convolutional neural network (CNN) capable of differentiating between mild and severe endoscopic disease, whilst Maeda et al. [254] developed an AI model using endocystopic images to predict histologic inflammation in UC. This model was able to identify persistent histologic inflammation with 74% sensitivity and 97% specificity and accurately detected histologic inflammation in 90% of cases with 100% inter-observer reproducibility [254]. A further study by Matalka et al. [255] has also demonstrated positive outcomes in AI-assisted histological assessment in UC, whereby their AI system was able to correctly classify 116 out of 118 biopsies as compared to the consensus of three expert pathologists. This expanding and evolving technology could become extremely valuable in assisting the established screening tools already in place, especially by aiding in the accurate and unbiased grading of inflammation. Furthermore, AI has the potential for application in assisting in the identification of dysplastic lesions (endoscopically and histologically), increasing inter-observer agreement without the need of additional physicians or more complicated scoring systems, and ultimately guiding subsequent surveillance strategies.

## 6. Conclusions

Predicting cancer occurrence and risk in patients with UC and CD, guiding the appropriate surveillance and intervening with the most fitting course of management is a very challenging process for this cohort of patients. The introduction of a surveillance programme, advances in medical care and endoscopic technology, recognising clinical risk factors strongly predictive of CA-CRC and tailoring surveillance accordingly has no doubt contributed to the falling incidence of this most feared complication in patients with IBD over the years. Despite this, current meta-analysis (81 studies and 181,923 patients) demonstrates an incidence rate of CA-CRC as 1.58 per 1000 patient-years (95% CI 1.39–1.76) [9].

This review has described the established risk factors that are currently utilised to identify IBD patients who are at high-risk of progression to CA-CRC. Undoubtably the clinical application of these has improved quality of life for many patients. We further highlight some of the molecular mechanisms used as predictors of CA-CRC, such as aneuploidy and field cancerisation, that could potentially be utilised more effectively for risk stratification in the surveillance programme. However, this review has also illustrated a lack of molecular understanding of the biology underlying cancer progression still remains, and this presents both a challenge and an opportunity for clinicians and scientists.

Our lack of molecular understanding means that there is room for improvement in risk stratification. On the one hand, failure to identify low-risk patients means they are at risk of over surveillance or management, leading to more invasive procedures, the potential for unnecessary or life-changing surgery, as well as the increased burden and cost of an over-used and over-stretched surveillance programme on health services. On the other hand, not identifying high-risk patients renders a patient susceptible to suboptimal or delayed management, with missed neoplasia and interval cancers still too frequent. However, overcoming these shortfalls, and channelling energy and resources towards understanding the biology of disease, with the potential of novel molecular biomarkers, both invasive and non-invasive (for example sWGS to measure aneuploidy, and cfDNA from blood samples), could present the opportunity to modify risk stratification to a more individualised approach.

By combining these established and evolving methods of surveillance with molecular biomarkers and emerging new directions, and incorporating an evolutionary analysis of the data, we have the potential to further optimise IBD surveillance. This will lead to improved clinical outcomes for patients and clinicians, as well as relieving an over-stretched health system compounded by a new COVID-era.

## Figures and Tables

**Figure 1 cancers-13-02908-f001:**
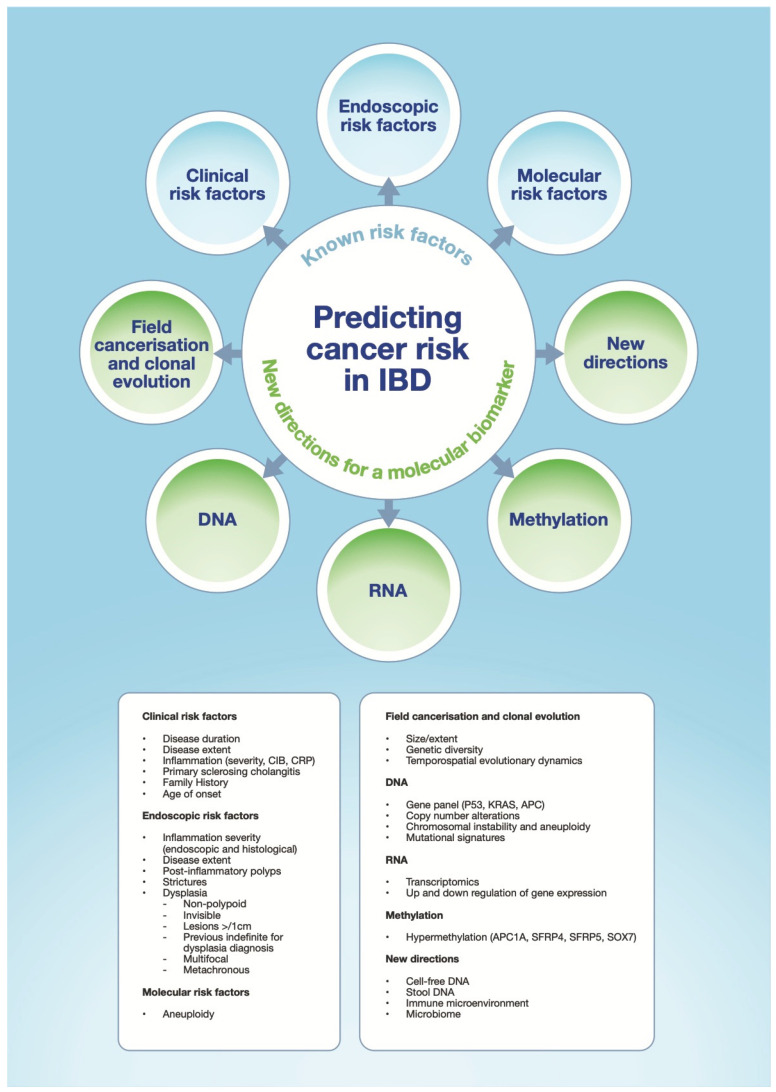
Demonstrates the known risk factors for predicting cancer risk in patients with IBD, and illustrates the potential novel directions for a molecular biomarker for disease progression. CIB—cumulative inflammatory burden, CRP—C-reactive protein.

**Table 2 cancers-13-02908-t002:** Summary of current IBD surveillance guidelines from European societies (BSG, ECCO and NICE).

**Risk factors**	Pancolitis with no active inflammation	Pancolitis with mildly active inflammation	Pancolitis with moderate-severe inflammation (endoscopic or histological),
	(endoscopic or histological),	(endoscopic or histological),	or dysplasia or strictures within past 5 years (±surgery)
	or left sided UC or CD of similar extent	or presence of post-inflammatory polyps,	or PSC,
	(i.e., <50% mucosa involved)	or family history of CRC in 1st degree relative > 50 yoa	or family history of CRC in 1st degree relative < 50 yoa
**Risk**	Low	Intermediate	High
**Surveillance**	5 year	3 year	Annual

(BSG—British Society of Gastroenterology, ECCO—European Crohn’s and Colitis Organisation, NICE—National Institute for Health and Care Excellence, yoa—years of age).

## Data Availability

MDPI Research Data Policies.

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
