# Peer review of "Predicting Colorectal Cancer Occurrence in IBD"

_cancers, 2021, doi:10.3390/cancers13122908_

Round 1

Reviewer 1 Report

The review by Yalchin et al provides a comprehensive in-depth overview of colitis associated CRC. It is well written and goes into depth in all aspects. I have only a few points to consider:

  1. The tile should be changed to ‘Predicting colorectal cancer occurrence in IBD’ as you do not look at non-CRC cancers.
  2. When referring to St Mark’s experience on page please highlight that this is a predominantly tertiary referral centre.
  3. RCT required to assess efficacy of colonoscopy surveillance: The real outcome should be avoidance of CRC related death not occurrence of dysplasia or cancer.
  4. Dysplasia 1.5 please split into visible and macroscopically not visible dysplasia. Please avoid the term LGD lesion as you also include flat LGD which is often not considered a lesion (ie non well circumscribed).
  5. The reason for lower CA-CRC rate sin Scandinavia may be the higher ate of colectomy (after which you won’t develop CRC).
  6. Biochemical assessment (2.1). Please clarify whether any associations between the parameters discussed relate to the value at CRC presentation or whether these are predictors of subsequent CRC development.

Author Response

RE: Revised manuscript cancers-1238990 “Predicting colorectal cancer occurrence in IBD

Dear Editors and Reviewers,

On behalf of my co-authors, thank you for taking the time to review our manuscript. We are pleased to submit a revised version for your approval. 

We thank the reviewers for their useful suggestions, and below we have provided a point-by-point response to each comment.

Reviewer 1

1. The tile should be changed to ‘Predicting colorectal cancer occurrence in IBD’ as you do not look at non-CRC cancers.

We have amended the title as suggested.

2. When referring to St Mark’s experience on page please highlight that this is a predominantly tertiary referral centre.

We have added this to the text (see page 2).

3. RCT required to assess efficacy of colonoscopy surveillance: The real outcome should be avoidance of CRC related death not occurrence of dysplasia or cancer.

We agree and have amended the text as suggested (see page 2).

4. Dysplasia 1.5 please split into visible and macroscopically not visible dysplasia. Please avoid the term LGD lesion as you also include flat LGD which is often not considered a lesion (ie non well circumscribed).

We thank the reviewer for highlighting this and have amended the text (see page 5).

5. The reason for lower CA-CRC rates in Scandinavia may be the higher ate of colectomy (after which you won’t develop CRC).

We have added this important point to the text (see page 2).

6. Biochemical assessment (2.1). Please clarify whether any associations between the parameters discussed relate to the value at CRC presentation or whether these are predictors of subsequent CRC development.

We apologise for the lack of clarity and have added the following text (see page 7): ‘Although, these studies suggest an association between higher levels of inflammatory markers and the presence of more advanced disease, their use as a predictive biomarker is still to be determined. RCT’s with long-term follow up are required to demonstrate the predictive value of these parameters for CA-CRC’.

Reviewer 2 Report

This review covers many aspects of cancer and risk in IBD

SPECIFIC COMMENTS

  1. IBD is written as Inflammatory bowel disease (only I is capitalised)
  2. The past part of the summary is vague
  3. The INTRO is very long
  4. Referencing needs correcting. When using et al the reference number should follow immediately (e.g. Eaden et al. [6]...)
  5. The term "UC patients" should be "patients with UC"
  6. some of the text could be reduced in length 

Author Response

RE: Revised manuscript cancers-1238990 “Predicting colorectal cancer occurrence in IBD

Dear Editors and Reviewers,

On behalf of my co-authors, thank you for taking the time to review our manuscript. We are pleased to submit a revised version for your approval. 

We thank the reviewers for their useful suggestions, and below we have provided a point-by-point response to each comment. Also, please see attachment. 

Reviewer 2

1. IBD is written as Inflammatory bowel disease (only I is capitalised)

We thank the reviewer for highlighting this error and have amended accordingly (see page 1).

2. The past part of the summary is vague

We apologise for lack of clarity here. We have added further detail, whilst keeping within the short word limit for this section.

3. The INTRO is very long

We have removed text from several sections of the introduction, notably content which is repeated later in the manuscript.

4. Referencing needs correcting. When using et al the reference number should follow immediately (e.g. Eaden et al. [6]...)

We apologise for this error and have amended across the entire manuscript.

5. The term "UC patients" should be "patients with UC"

We have amended this as suggested across the entire manuscript.

6. some of the text could be reduced in length 

We thank the reviewer for highlighting this and we have reduced the length of text where appropriate, notably on page 8 (2.3) and page 17 (4.3 and 4.4).

We hope that you will find that these alterations address the reviewer’s comments in full, however please let me know if you would recommend any further amendments. Thank you for your consideration of this manuscript for publication in the Special Issue of Cancers Journal.

Sincerely

Dr Mehmet Yalchin, BSc, MBBS, MRCP

Clinical Research Fellow Gastroenterology

St. Marks’s Hospital
